# A New Method of Identifying Characteristic Points in the Impedance Cardiography Signal Based on Empirical Mode Decomposition

**DOI:** 10.3390/s23020675

**Published:** 2023-01-06

**Authors:** Paulina Trybek, Ewelina Sobotnicka, Agata Wawrzkiewicz-Jałowiecka, Łukasz Machura, Daniel Feige, Aleksander Sobotnicki, Monika Richter-Laskowska

**Affiliations:** 1Institute of Physics, Faculty of Science and Technology, University of Silesia in Katowice, 41-500 Chorzow, Poland; 2Łukasiewicz Research Network—Krakow Institute of Technology, The Centre for Biomedical Engineering, Zakopianska Str. 73, 30-418 Krakow, Poland; 3Department of Physical Chemistry and Technology of Polymers, Silesian University of Technology, 44-100 Gliwice, Poland; 4PhD School, Silesian University of Technology, 2A Akademicka, 44-100 Gliwice, Poland

**Keywords:** ICG, fiducial points, characteristic points, impedance cardiography, EMD, EEMD, stroke volume

## Abstract

The accurate detection of fiducial points in the impedance cardiography signal (ICG) has a decisive impact on the proper estimation of diagnostic parameters such as stroke volume or cardiac output. It is, therefore, necessary to find an algorithm that is able to assess their positions with great precision. The solution to this problem is, however, quite challenging with regard to the high sensitivity of the ICG technique to the noise and varying morphology of the acquired signals. The aim of this study is to propose a novel method that allows us to overcome these limitations. The developed algorithm is based on Empirical Mode Decomposition (EMD)—an effective technique for processing and analyzing various types of non-stationary signals. We find high correlations between the results obtained from the algorithm and annotated by an expert. This, in turn, implies that the difference in estimation of the diagnostic-relevant parameters is small, which suggests that the method can automatically provide precise clinical information.

## 1. Introduction

Impedance cardiography (ICG) is a technique that measures the changes in impedance across the thorax in a non-invasive way and can be used to monitor important hemodynamic parameters without possible risks to or complications regarding a patient’s health. The usefulness of this technique in the context of hemodynamic parameters was introduced and confirmed in 1978 by Miller et al. [1]. Despite the non-invasive, inexpensive and easy-to-use characteristics of ICG, this technique is not frequently applied in clinical practice [2]. On the one hand, this is caused by the ICG’s limitations, i.e., patient obesity and restriction of body weight (25–250 kg), heart rate limit up to 250/min, hyperactivity during the measurement, or incidents of arrhythmia [3,4,5]. In addition, the complex character of ICG signals pose some difficulties in the proper identification of characteristic points and further estimation of the hemodynamic parameters. There are several algorithms and methods of ICG signal analysis that allow for the determination of characteristic points, which are directly translated into valuable hemodynamic features, including left ventricular ejection (LVET), stroke volume (SV), or cardiac output (CO).

The identification of the most important ICG signals points is related to the phase of the cardiac cycles. The approaches to ICG point characterization have changed over the years in order to increase their estimation stability. However, the accurate detection of specific ICG points, which are clearly described in a forthcoming section, is quite challenging. To start with, we have to be very careful with the filtering process—due to the heart rate variability, some techniques may blur less distinctive events, making the proper identification of points impossible. Second, the problem also comes from the changing morphology of dZ/dt waveforms [6,7]. Eventually, there exists no uniform definition of the points marking the aortic valve opening and closing (B, X), which leads to ambiguities in the points identification even by expert cardiologists.

There are different approaches in the literature to overcoming the abovementioned problems. The authors of [8] distinguish three kinds of techniques, detecting the characteristic points in the ICG signal. Most algorithms use the positions of the R peak in the ECG signal in order to estimate the interval in which the C point is searched for [9,10,11,12]. Then, C point is established as the highest point occurring in this interval. Based on its position, other points are identified with various mathematical conditions involving extrema, zero-crossings, and higher-order derivatives [13,14,15,16,17]. Another group of methods utilizes the time–frequency distribution of the signal [18] and the wavelet decomposition. Within these approaches, the characteristic points are either identified as the singularities of the wavelet transform [19,20,21,22] or inferred from the distribution of wavelet coefficients [23,24]. The majority of them struggle, however, with a high computational complexity or a large sensitivity to the artifacts.

In this article, we present a new approach to the estimation of characteristic points of an ICG signal using the modified version of the empirical mode decomposition method (EMD). The EMD is an empirical technique that decomposes the signal into a finite number of frequency components. A significant advantage of this technique lies in the fact that, in contrast to the other commonly used frequency-based techniques, i.e., Fourier transform, the EMD can be successfully applied to the signals that characterize the non-stationary and nonlinear processes. Signal decomposition into frequency components is used in many fields of electrophysiological time series analysis, including the de-noising procedure [25,26,27], the removal of a signal’s trend [28,29], the identification of the most valuable components, and further statistical analysis [30]. One of the most common ways to use EMD is the EMD-based classification of time series and features extraction. Based on the features calculated for the individual frequency components, the further classification is performed with standard methods of classification or machine learning approaches [31,32].

In the context of a biomedical signal analysis, the EMD techniques and their different variants are widely applied in the analysis of ECG. It has been demonstrated that this method can be helpful in tracking different cardiac arrythmia [33,34], in properly detecting characteristic points of the signal [35,36,37,38,39,40,41], in T-wave alternates [42], and in signal processing [25,40,43,44,45,46,47,48,49]. However, much less attention in the literature has been paid to the possible applications of this algorithm in the impedance cardiography signal. We found only a few reports regarding the usage of EMD in its analysis. In [50,51,52], it was shown that EMD can be extremely useful in the identification of cardiovascular diseases. It also facilitates the detection of characteristic points of this signal [53] and allows for effectively suppressing noise without losing significant features in the acquired recordings [54,55]. Here, we propose a novel, EMD-based method that allows for the accurate detection of fiducial points in the ICG signal. We find that the positions of the characteristic points identified by this algorithm are in good agreement with those annotated by an expert cardiologist, which results in accurate assessments of the hemodynamic parameters and facilitates the diagnostic process. In addition, due to the low computational complexity of this algorithm and the lack of requirements for simultaneous recording of the ECG signal, the proposed method can be implemented on ultra-low power devices used in the remote monitoring of cardiac patients. The paper is organized as follows. In Section 2, we present a brief introduction to the method of impedance cardiography and provide a detailed description of the applied algorithms: EMD (empirical mode decomposition) and EEMD (ensemble empirical mode decomposition). In Section 3, we introduce a new method identifying ICG characteristic points based on the previously mentioned EMD and EEMD techniques. We also study the accuracy of the proposed algorithm in predicting the values of the hemodynamic parameters. Section 4 and Section 5 contain a discussion of the results and the final conclusions.

## 2. Methods

### 2.1. Impedance Cardiography

The bioimpedance method, also called impedance rheography, is a diagnostic technique for assessing the function of the body’s internal organs on the basis of the impedance value or its changes in the examined area of the body. Impedance fluctuations occurring as a result of changes in the volume and velocity of the blood vessels, air in the lungs, as well as movements of organs and changes in their shape can provide information about the state and function of the body’s internal organs [4,56,57,58]. The undoubted advantage of the bioimpedance method is its non-invasiveness and lack of known side effects [59]. During the tests, electrical impedance measurements are made of the tested tissue area. The examination consists of the application of an alternating application current of constant amplitude into a tested area. Then, voltage recordings allow for continuous measurement of the impedance values. The measured impedance depends on the blood volume in the tested area, temperature, and blood resistivity, affected by spatial orientation of erythrocytes and by hematocrit (HCT) [57,60].

The relationships between changes in blood volume and the impedance of a given body segment are described using simplified models, such as the cylindrical model, the Cole–Cole model, or the model by Hanai [61]. These models provide the basis for an analytical, mathematical description of the attenuation and electrical permeability of biological tissues and form the basis of impedance cardiography. In particular, they allow for reliable and non-invasive determination of the cardiac stroke volume (SV, which is considered the most important measure of the mechanical work behind the heart [56,57,62]. According to the formula developed by Kubicek and his coworkers, SV can be calculated as follows [63]:(1)$$\mathrm{SV}=\frac{\rho L^{2}}{Z_{0}^{2}}{\left(\frac{dZ}{dt}\right)}_{\max}\cdot \mathrm{LVET},$$
where ρ is the specific resistance of blood, *L* denotes the thoracic length between voltage electrodes, $Z_{0}$ is a base impedance, ${\left(\frac{dZ}{dt}\right)}_{\max}$ corresponds to the maximum rate of a change in bioimpedance, and LVET is a ventricular ejection time.

The expression given in Equation (Equation 1) was developed on the basis of the cylindrical model of the thorax [56,58,64] presented in Figure 1.

The model assumes that the resulting waveform of impedance changes ΔZ is due only to volume changes in the heart and large blood vessels and that the resistivity of blood during flow is constant [56]. Figure 2 presents an example of the impedance change curve REO, as well as the derivative of the bioimpedance curve ICG for a single cardiac cycle.

The values of ${\left(\frac{dZ}{dt}\right)}_{\max}$ and LVET necessary to estimate the stroke volume SV can be determined based on the positions of the characteristic points B, C, and X located on the first derivative of the bioimpedance curve (ICG) [58].

Each of these points has a clear interpretation. Point B corresponds to the moment of the aortic valve opening, point C denotes a maximum blood ejection speed, while point X indicates the moment of the aortic valve closing [58]. A brief description of these fiducial points along with the hemodynamic parameters that are determined based on their locations in the ICG signal are introduced in Table 1. Some of the most important hemodynamic quantities are CO, PEP, LVET, and the Heather Index. Their definitions are presented in Table 2.

### 2.2. Empirical Mode Decomposition (EMD)

#### 2.2.1. Standard EMD

Empirical mode decomposition (EMD) is a technique that decomposes the signal X(t) into a finite number of modes, referred to as intrinsic mode functions (IMFs), and the residuum signal $r_{n}\left(t\right)$:(2)$$x\left(t\right)=\sum_{i=1}^{n}C_{i}\left(t\right)+r_{n}\left(t\right)$$
where $C_{i}\left(t\right)$ stands for an IMF sequence and $r_{n}\left(t\right)$ is the residual signal.

Modes are determined by locally dominant frequencies. Worth emphasizing is that EMD has the ability to process the nonlinear and non-stationary time series, in contrast to other commonly used techniques, i.e., the Fourier Transform. This property is directly related to the empirical nature of the EMD technique. In other words, the data itself dictates the decomposition and there is no prescribed analytical formula, which is the essence of the algorithm. The method was originally developed by Huang et al. in 1998 [65]. In its standard, it contains the following steps:1.First, the maxima and minima of the sequence are identified.2.Next, the upper and lower envelopes over found extremes are constructed by cubic spline line interpolation.3.The mean value m(t) of the upper and lower envelopes is estimated. This average value is subtracted from the original series, h(t)=X(t)−m(t), and the resulting signal h(t) is treated as the potential candidate to be the first IMF. Each IMF must fulfill two conditions:The number of extrema (maxima and minima) and the number of zero crossings must be equal to or differ at most by one;The average value of the upper and lower envelopes defined by local maxima and minima must be zero.4.If the conditions are not fulfilled, the procedure is repeated starting from step 1, this time with h(t) as an input signal.5.After the identification of the first IMF, this component is subtracted from the input series. Then, the obtained residuum must meet the following stopping criterion:The residuum signal has only one extremum (minimum or maximum) or is represented by a constant/monotonic function. That kind of residuum signal characterizes the trend of the time series.6.The whole procedure of this sifting process ends when a residual signal is found. If the criterion of being residual sequences is not fulfilled yet, the procedure is repeated from step 1 with a residuum as an input series. A diagram of the main standard EMD stages is presented in Figure 3.

#### 2.2.2. Ensemble Empirical Mode Decomposition

The standard version of EMD faces some difficulties, such as the mode mixing problem, which occurs when one frequency is not assigned to the one mode but is spread over several components. It leads to frequency mixing between different modes and, as a result, to an inefficient decomposition. To overcome this problem, modified versions of the EMD, such as ensemble empirical mode decomposition (EEMD) [66] or, more recently, complete mode empirical decomposition (CEEMD) [67], have been proposed.

The first one—EEMD—involves EMD of the original signal slightly perturbed by a Gaussian noise.

The EEMD procedure can be summarized in the following steps:1.A white noise sequence w(t) is added to the time series under consideration xw(t)=x(t)+w(t).2.The noisy signal w(t) is decomposed into IMFs through the standard procedure of empirical mode decomposition described in the previous subsection.3.The first two steps (1 and 2) are repeated for $N_{t}$ different realizations of white noise.4.EEMD-based IMFs are estimated by averaging the ensemble of IMFs:
(3)$$IMF_{l}\left(t\right)=\frac{1}{N_{t}}\sum_{i=1}^{N_{t}}IMF_{l}^{i}\left(t\right),$$
where $IMF_{l}\left(t\right)$ is the l−th-order IMF and *i* stands for the number of realizations.

## 3. Results

### 3.1. New Method of ICG Characteristic Point Identification

The proposed method allows one to evaluate the positions of the most crucial points: B, C, and X used during the assessment of the hemodynamic parameters. Here, we assume that the signal has already been subjected to an initial preprocessing stage. Then, we start from the detection of the C point. Its accurate assessment is crucial for evaluating the heart rate value. The position of this point is also often treated as the prerequisite for the evaluation of the other points (B and X), which have a decisive impact on the values of the hemodynamic parameters.

Within the proposed approach, the first step taken towards the detection of the C peaks is EMD of the registered signal after initial pre-filtering. As a result, we obtain several IMFs (their number can vary between different traces), as depicted in Figure 4. As C point is the most prominent point appearing in the cardiac cycle, it can be extracted from the lower-order IMFs. From Figure 4, one can infer that only the first four IMFs influence the position of this point. Then, the function $cf_{11}$, defined as
(4)$$cf_{11}=\sum_{i=1}^{4}IMF_{i},$$
contains all necessary information to accurately estimate it. However, with regard to the high resemblance of $cf_{11}$ to the original signal containing a lot of fluctuations, estimation of the C point based on the $cf_{11}$ function alone, is quite challenging (see Figure 5). In order to suppress fast oscillations and to make C point more pronounced, we multiply the function $cf_{11}$ by $cf_{12}$:(5)$$cf_{12}=\prod_{i=1}^{3}IMF_{i}.$$

Then, by taking the absolute value of this product,
(6)$$cf_{1}=\left|cf_{11}\cdot cf_{12}\right|$$
we obtain the signal in which the C peaks are easily detectable (see Figure 6). They are associated with the points of the largest amplitudes lying above the threshold Th updated in each iteration according to the algorithm described in [68]. Following [69], we also assume a refractory period of 200 ms between two consecutive searches.

As for the B point, corresponding to the moment of aortic valve opening, it can be identified as the first maximum preceding C point in the first derivative of the fourth-order component $IMF4^{\prime }$ obtained from the EEMD (see Figure 7).

When it comes to the X point, the lower-order components of the EEMD are responsible for its position. Based on the empirical observations, we found out that following combination of IMF3, IMF4, and IMF5 is the most useful in its identification:(7)$$cf_{2}=\mathrm{IMF}3+2\cdot \mathrm{IMF}4+4\cdot \mathrm{IMF}5.$$

The X point is searched for within the interval ${\mathrm{Int}}_{X}$:(8)$${\mathrm{Int}}_{X}=\left[C,C+0.15\cdot CC_{\mathrm{mean}}\right],$$
where *C* is the position of the C point in the current cycle and $CC_{\mathrm{mean}}$ denotes the mean difference between two consecutive C points (mean heart rate). The $cf_{2}$ function along with the marked positions of X points identified by the algorithm is depicted in the Figure 8.

The above described method of B, C, and X point identification is summarized in Table 3. We also present the main flowchart of the proposed algorithm in Figure 9.

### 3.2. Test of the Algorithm

#### 3.2.1. The Database

In order to assess the efficiency of the developed method, we used a publicly available database designed for the purpose of testing ICG delineation algorithms [70]. This database contains 48 impedance cardiography signals from 24 healthy subjects, recorded during an experimental session of a virtual search and rescue mission with drones [70,71]. For each subject, two 5 min signals corresponding to the baseline (registered during a passive task) and cognitive workload (registered during a task demanding large mental effort) states were obtained. Since data acquisition from rescuers during real missions is quite challenging, signals are gathered during virtual sessions simulating different levels of cognitive overload. During these sessions, a pilot needs to fly a drone following the provided pathway and indicate a disaster area. All physiological recordings were acquired with the Biopac: MP160 Data Acquisition Systems (Biopac Systems Inc., Goleta, CA, USA).

All acquired traces include beat-to-beat annotations of the ICG characteristic points, performed by an expert cardiologist. In addition, it contains, as a reference, synchronously registered ECG signals, which facilitate the precise annotation of the cardiac events. Apart from the raw data, signals subjected to a filtering process are also provided. The prepossessing step, which is applied here, consists of two parts. At the beginning, the registered traces are downsampled from 2000 Hz to 250 Hz. The remaining artifacts are then suppressed with the adaptive Savitzky–Golay filter of order 3 [72,73]. It turns out that it yields a good balance between the signal de-noising and maintenance of its characteristic features [74]. Then, for each cardiac cycle, four characteristic points are indicated by an expert cardiologist: B, C, X, and O.

#### 3.2.2. Algorithm vs. Expert—Statistical Analysis of the Emerging Differences

The TIBCO$Statistica^{TM}$ statistical package in version of 14.0.0.15 together with the open source libraries dedicated to Python were used for the data analysis. The hypothesis about the normal distribution of analyzed variables was verified via the Shapiro–Wilk formula at the significance level α=0.05. For the investigation of the statistical significance of differences, the t-test and its non-parametric equivalent in the form of the Mann–Whitney U statistic were used. Figure 10 summarizes the statistical comparison of the hemodynamic parameters obtained by the algorithm and marked by an expert in cardiology. For the vast majority of cases, the determined p-values calculated for the Shapiro–Wilk test were less than the assumed significance level α=0.05. For that reason, box–whisker plots characterizing the respective quartiles are presented instead of the average values with the standard deviations. The subplots compare the median values of selected hemodynamic parameters obtained by the algorithm and the expert. Although the changes between respective groups (expert vs algorithm) are relatively small, according to the Mann–Whitney U formula, there exist statistically significant differences between the compared cases at the selected significance level α=0.05 (the *p*-values are highlighted in each subplot). The values of selected hemodynamic parameters determined by the EMD-based algorithm are slightly overestimated compared to the results indicated by the expert. Table 4 characterizes the differences between median values of the compared parameters estimated by the algorithm and an expert. To assess what kind of characteristic points could mostly affect the differences between the algorithm and the expert, the accuracy values were calculated and presented in Table 5. Following [74], ±30ms is considered the tolerance between positions of points annotated by the expert and those identified by the algorithm. A good agreement can be observed for the B and C points (significantly above 90%). Simultaneously, our results suggest that X is the most difficult point to determine automatically. Compared to other algorithms of characteristic point determination in ICG signals, the agreement with the expert is also the smallest in the case of the X point.

## 4. Discussion

This article proposes a new approach to determining characteristic points in the ICG signal. Several algorithms for determining ICG points are described in the literature. However, there is no comprehensive information on their effectiveness in the estimation of important hemodynamics features, and it is difficult to decide what kind of method is the gold standard for ICG signal characterization. Some of them rely on higher-order derivatives, which are very noise-sensitive. Others are based on R peak detection, which requires simultaneous recording of the ECG signal. Eventually, there are also algorithms based on the wavelet decomposition. Although their efficiency is quite high, they are also computationally expensive. Here, we propose a method that does not involve additional ECG signal analysis and is fast and accurate.

It turns out that decomposition of a signal into individual frequency modes using an empirical technique can be a very good basis for the automatic localization of the most important points in an ICG signal. Our results suggest that the selection of specific frequency components or the combination of these components can greatly assist in identifying the correct location of the desired points. In the majority of cases, the estimated compatibility of an EMD-based methodology between the expert and algorithm was within or above 90%. It is worth mentioning that a high degree of agreement was obtained both for measurements in the resting (reference) state and in the case of stressors dictated by a specific task. Only in the case of the X point was there a slight decrease below the ninety percent threshold. Note that the lower accuracy of this point identification in comparison to other methods stems from the more strict value of the permissible tolerance between an expert and the algorithm: (±30 ms) in this work vs. (±150 ms) in [9,75,76]. Discrepancies between an Expert and the Algorithm are described in Appendix A.

The statistically significant differences between an expert and an algorithm were identified for all the analyzed parameters. However, the differences between the respective median values are relatively small and may be directly related to many factors, including sensitivity of the results to even small perturbations in the recording, which occur in the case of a measurement during an activity. It should be emphasized that the demonstrated significance of the differences does not necessarily translate into their physiological meaning. For example, in the case of the LVET parameter, the identified difference between the EMD-based algorithm and the expert is about 20 ms, while the permissible difference is 150 ms. In addition, some publications allow even larger expert–algorithm dissimilarity. The proposed characteristic point extraction technique requires further improvements to overcome some shortcomings related primarily to the determination of the X point. The key to achieving better accuracy may be the inclusion of the removal of noise during data preprocessing. Nevertheless, the simplicity, speed of operation of the algorithm, and its empirical nature seem to be the overarching advantages in the context of the need to automate the process of determining ICG characteristic points. In terms of the accuracy of identifying the key ICG points, our results are comparable to the results obtained by the other algorithms presented in the literature, and in some cases, they show even greater precision. In the work by Pale et al. [74], the authors presented a methodology that allows one to obtain accuracies of about 94.9%, 98.6%, and 90.3%, respectively, for the B, C, and X points, comparing the output of the algorithm with the manual identification of the crucial points by the cardiologist. In our case, the values are, respectively, 96.9%(B), 98.1%(C), and 88.6%(X). It is worth emphasizing that, in the literature, the margin of error is considered to be about 150 ms, while in this paper, we have limited it to only 30 ms.

## 5. Conclusions

Despite the importance of the ICG technique in different healthcare applications, the analysis and classification of ICG signals by applying advanced signal processing techniques is still very limited. A new approach to identifying characteristic points (B, C, and X) on the impedance cardiograph signal developed in this work based on ensemble empirical mode decomposition is very promising. The algorithm is fast, does not require simultaneous recording of the ECG signal, and yields quite accurate results. However, there is still much room for improvement, especially in terms of the appropriate filtering process, and the proposed algorithm has the potential to be implemented on low-power electronic devices remotely monitoring the conditions of cardiac patients.

## Figures and Tables

**Figure 1 sensors-23-00675-f001:**
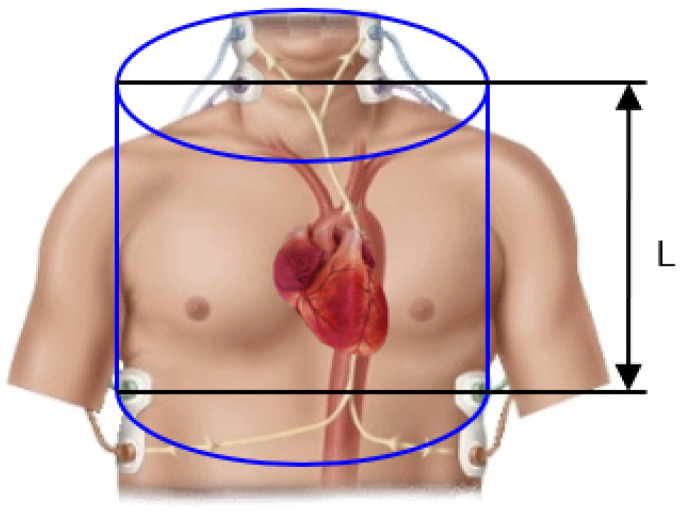
Cylindrical chest model according to Kubicek. *L* corresponds to the length between voltage electrodes applied during the ICG measurement.

**Figure 2 sensors-23-00675-f002:**
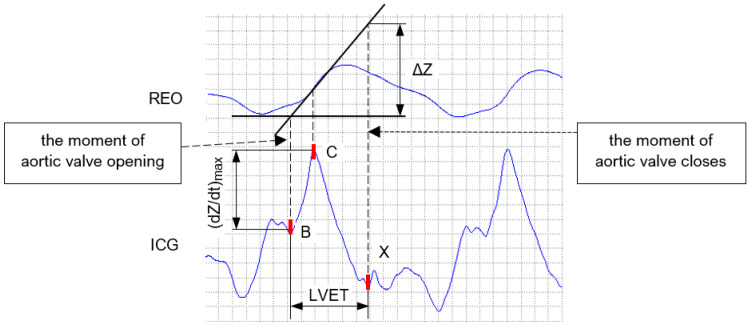
The way to determine a change in the bioimpedance during one cardiac cycle [58].

**Figure 3 sensors-23-00675-f003:**
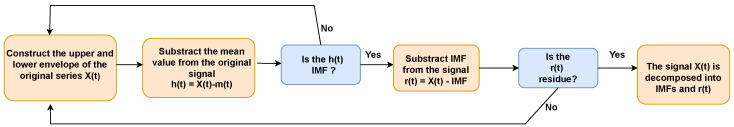
Illustration of the main stages of standard empirical mode decomposition.

**Figure 4 sensors-23-00675-f004:**
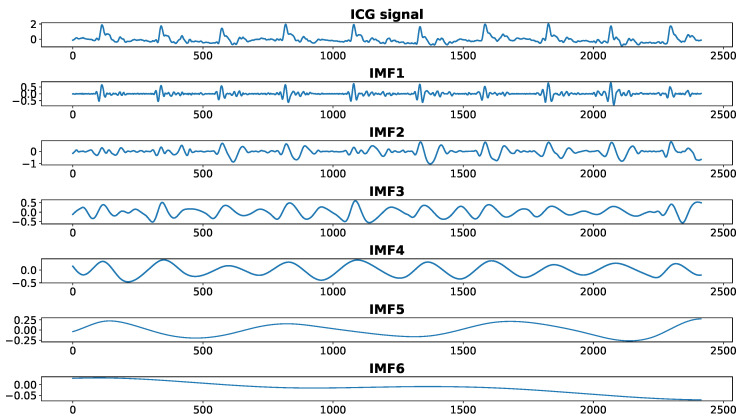
First 6 IMFs obtained from EMD of an ICG recording.

**Figure 5 sensors-23-00675-f005:**
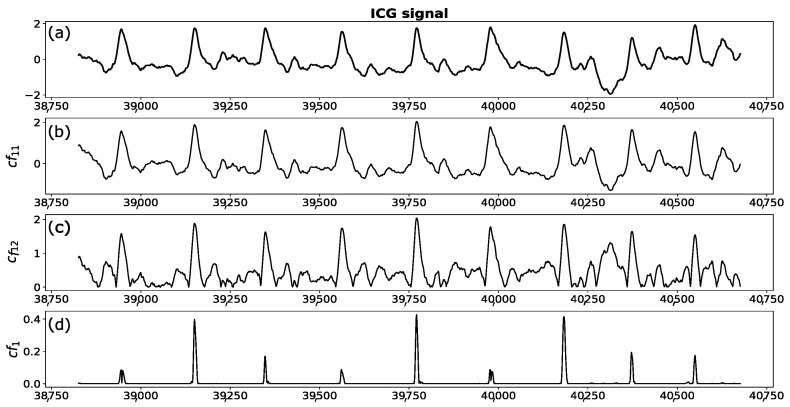
The upper figure (**a**) represents a fragment of an ICG signal. Panel (**b**) illustrates the function $cf_{11}$, representing the sum of the first four intrinsic mode functions (IMFs) resulting from EMD of the signal presented in panel (**a**). Panel (**c**) shows the function $cf_{12}$ defined as the product of first three IMFs. In the lower panel (**d**), the product of the signals (**b**,**c**) is depicted.

**Figure 6 sensors-23-00675-f006:**
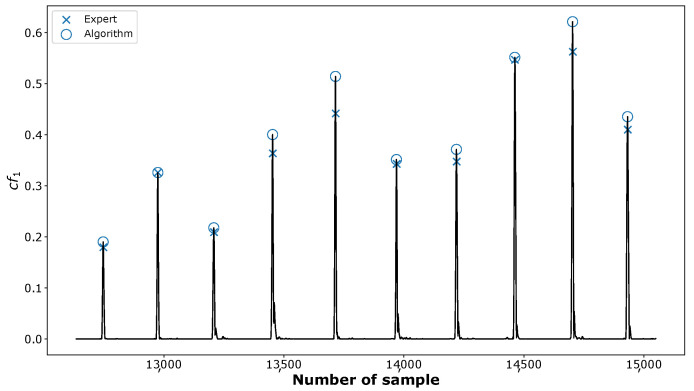
The positions of the C peaks marked in the combination of IMFs $cf_{1}$ identified by an expert (crosses “×”) and detected by the proposed algorithm (circles “o”).

**Figure 7 sensors-23-00675-f007:**
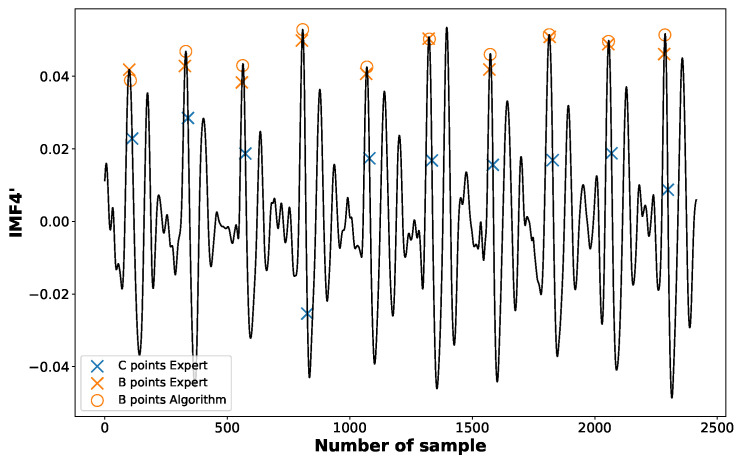
The positions of B points marked according to an expert (crosses “×”) and identified by the proposed algorithm (circles “o”).

**Figure 8 sensors-23-00675-f008:**
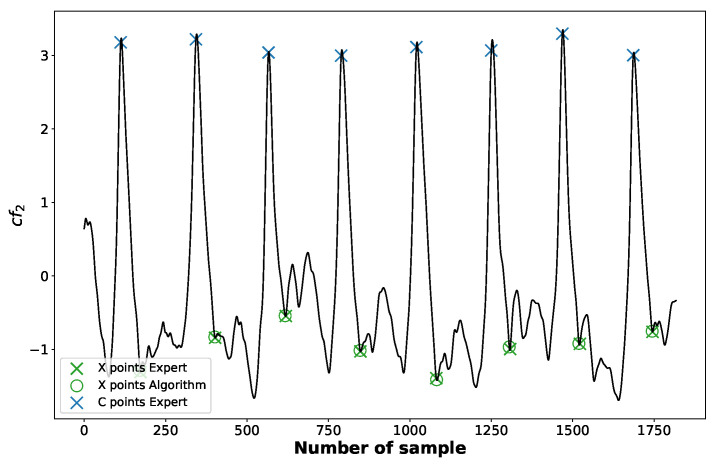
The positions of X points identified by an expert (crosses “×”) and those detected by an algorithm (circles “o”).

**Figure 9 sensors-23-00675-f009:**
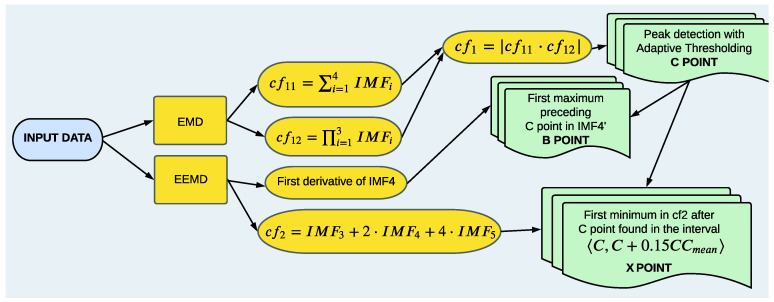
A flowchart illustrating the working principle of the proposed algorithm. The input data are decomposed with the EMD and EEMD methods. Then, functions $cf_{11},\;cf_{12},\;\mathrm{IMF}4^{\prime },\;cf_{2}$ of the obtained IMFs are used to find the positions of the C, B, and X points.

**Figure 10 sensors-23-00675-f010:**
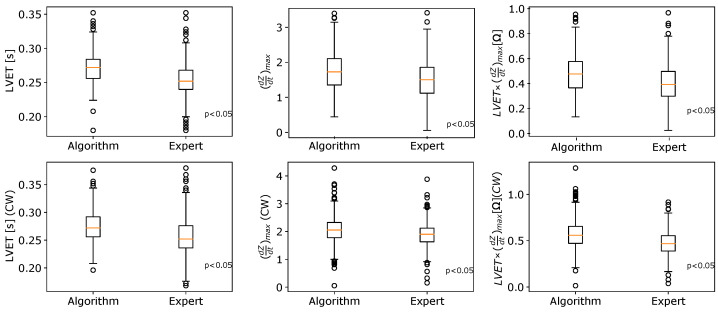
The box-plots comparison of hemodynamic parameters calculated by the algorithm and indicated by an expert.

**Table 1 sensors-23-00675-t001:** A brief overview of the characteristic points of the ICG signal. The last column represents the hemodynamic parameters (described in more detail in Table 2), the values of which are influenced by their positions in the impedance cardiogram.

The Point	Description	Hemodynamic Parameters
**B**	The onset of rapid upstroke towards the C point. It represents the moment of aortic valve opening.	PEP, LVET, SV, CO
**C**	Point with the greatest amplitude in one cardiac cycle. It represents the maximum aortic flow.	HR, SV, CO, Heather Index
**X**	The minimum ICG signal in one cardiac cycle. It represents the moment of aortic valve closing.	LVET, SV, CO

**Table 2 sensors-23-00675-t002:** A brief description of the most popular hemodynamic parameters inferred from the positions of the characteristic points in the ECG and ICG signals.

The Hemodynamic Parameter	Definition
**PEP** (Pre-ejection period)	The time between electrical systole (Q point in ECG) and opening of the aortic valve (B point in ICG).
**LVET** (Left Ventricular Ejection Time)	The period of blood flow across the aortic valve. The time between B and X points in the ICG signal.
**HR** (Heart Rate)	The frequency of the heartbeat. The mean number of C points occurrences in one minute.
**Heather Index**	Cardiac contractility index defined as C/(C−Q).
**SV** (Stroke Volume)	Amount of blood ejected from the left ventricle during one cycle
**CO** (Cardiac Output)	Amount of blood ejected from the left ventricle in one minute

**Table 3 sensors-23-00675-t003:** Methods of identifying characteristic points in the ICG signal.

The Characteristic Point	EMD/EEMD Components	Method of Identification
**C point**	EMD 1, 2, 3, 4	Point of the largest amplitude occurring in the combinations of EMD lower–order components ($cf_{1}$ function).
**B point**	EEMD 4	First maximum preceeding C point in the first derivative of the fourth component $IMF4^{\prime }$ obtained from EEMD
**X point**	EEMD 3, 4, 5	First minimum after C point found in the combinations of EEMD higher–order components

**Table 4 sensors-23-00675-t004:** The median values of hemodynamic parameters estimated by the algorithm and an expert, and the respective differences between those two groups.

Parameter	Algorithm	Expert	$\Delta_{\mathrm{ALG}.-\mathrm{EXP}.}$
LVET	0.272	0.252	0.020
$\frac{dZ}{dt}\left(max\right)$	1.727	1.500	0.226
LVET × $\frac{dZ}{dt}\left(max\right)$	0.465	0.382	0.083
LVET (CW)	0.272	0.252	0.020
$\frac{dZ}{dt}\left(max\right)$ (CW)	2.053	1.920	0.151
LVET × $\frac{dZ}{dt}\left(max\right)$ (CW)	0.558	0.468	0.091

**Table 5 sensors-23-00675-t005:** The accuracy of characteristic point estimation via EMD-based algorithm. The comparison of the location of points estimated by an expert with the values determined by formulas based on IFMs. As matched locations for the expert and the algorithm, values differing by a maximum of ±7 points—approximately 30 milliseconds—were adopted.

Point	Number of Well Predicted Points	Number of All Points	Percentage Accuracy
B	775	779	96.92
B_CW	584	623	93.74
C	764	799	98.07
C_CW	610	623	97.91
X	690	799	88.58
X_CW	519	623	83.31

## Data Availability

The dataset used in this study is publicly available and can be found in [70].

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
