# Peer review of "A New Method of Identifying Characteristic Points in the Impedance Cardiography Signal Based on Empirical Mode Decomposition"

_sensors, 2023, doi:10.3390/s23020675_

Round 1
Reviewer 1 Report
1- At the end of the introduction, the next parts of the paper should be explained.
2- What is variable Nt in formula 3?
3- The changes of parameter cf12 should be shown in Figure 4.
4- In the caption of Figure 7, it is stated that the estimation of point X is marked with x, but there is another expression in the legend of figure.
5- It is necessary to compare the results with a similar study.
Author Response
Dear Reviewer,
We are thankful for the time devoted as well as all your professional remarks. We addressed every raised question and corrected the manuscript accordingly. Below we present our point-by-point response to the Reviewers’ comments.
- At the end of the introduction, the next parts of the paper should be explained.
Thank you for this remark. We included it at the end of the introduction, as suggested.
- What is variable Nt in formula 3?
Nt was defined in the line 153 (line above the formula) as the number of different realizations of white noise.
- The changes of parameter cf12 should be shown in Figure 4.
Thank the Reviewer for this tip. We added this function in this Figure (please see now Figure 5).
- In the caption of Figure 7, it is stated that the estimation of point X is marked with x, but there is another expression in the legend of figure.
Thank you for noticing this oversight. We changed the description in the legend of this Figure.
- It is necessary to compare the results with a similar study.
There is a lack of articles that takes into consideration the implementation of EMD technique in the ICG signal in order to identify characteristic points. Nevertheless, in response to the Reviewer suggestion we decided to compare our results with the results characterized by Pale et. al [74], where the authors presented the accuracy of the developed algorithm and compared it with the manual identification of B, C, X points by the cardiologist. This issue is now described in the last paragraph of the Discussion section.
Reviewer 2 Report
The submission “A new method of the identification of the characteristic points in the impedance cardiography signal based on the Empirical Mode Decomposition.” requires more attention to be ready for the publishing process:
The main problem statement requires more details.
The author needs to include a state-of-the-art literature review.
The author should tabulate the used datasets and provide an image that demonstrates the data.
In section 2, the proposed algorithm is not clear; the author needs to separate the background theory and proposed work in clear titles.
It is required to provide the main flowchart that describes the proposed algorithm.
The initial conditions or study setups are missed; the author needs to provide them.
Justify the use of 6 EMD stages in the proposed algorithm.
Add the results of all decomposite EMD output (not just IMF4) in section 3.
The fiducial points requires issue require more description and many important details are missed like how many points the algorithm targeted to figure out, the points distribution, which part of signal is most important .. etc.
What is the “Statistica 14.0 package”? What are the statistics that are employed in section 3.2.2? clarify them.
The author needs to illustrate the criteria that used in section 3.2.2. (algorithm vs expert).
Include the benchmarking table in the results section.
Author Response
Dear Reviewer,
We are thankful for the time devoted as well as all your professional remarks. We addressed every raised question and corrected the manuscript accordingly. Below we present our point-by-point response to the Reviewers’ comments.
The author needs to include a state-of-the-art literature review.
We included a short description in the introduction of the other algorithms detecting characteristic points in the ICG signal. We also gave a reference to the more detailed review regarding these algorithms and preprocessing methods.
The author should tabulate the used datasets and provide an image that demonstrates the data.
The data used comes from an external database available at https://doi.org/10.5281/zenodo.4725433 which was mentioned in the Data Availability Statement. The data has been characterized in the subsection 3.2.1. The raw input data were presented in many places within the manuscript. According to the Reviewer suggestion, we have made every effort (also through adding the additional diagrams) to describe and present the data in more detail.
In section 2, the proposed algorithm is not clear; the author needs to separate the background theory and proposed work in clear titles.
There is no description of the proposed algorithm in Section 2 - it is given in Section 3. Section 2 (Methods) introduces the background theory of impedance cardiography and two advanced methods of signal processing such as EMD and EEMD on which we built our concept.
It is required to provide the main flowchart that describes the proposed algorithm.
We included the flowchart describing the algorithm as suggested.
The initial conditions or study setups are missed; the author needs to provide them.
We added a more detailed description of the experimental setup in the subsection (“Database”).
Justify the use of 6 EMD stages in the proposed algorithm.
We present the most important stages of the EMD algorithm in, which characterize the main idea of the method. For the better representation of the iterative algorithm we add the diagram of individual steps of standard EMD (see Figure 3).
Add the results of all decomposite EMD output (not just IMF4) in section 3.
All intrinsic functions have been already depicted in Fig.3 (Section 2). In our opinion, presentation of this figure in the two sections is redundant and does not contribute to the better understanding of the algorithm.
The fiducial points issue requires more description and many important details are missed like how many points the algorithm targeted to figure out, the points distribution, which part of signal is most important .. etc.
The numbers of analyzed points are presented in Tab. 5.
What is the “Statistica 14.0 package”? What are the statistics that are employed in section 3.2.2? clarify them.
We have clarified the information about the statistical package Statistica 14. All the statistics employed in the form of the Shapiro-Wilk formula for testing normality, and the non-parametric tests used were also characterized more precisely.
The author needs to illustrate the criteria that are used in section 3.2.2. (algorithm vs expert). Include the benchmarking table in the results section.
Table 4 includes the median values presented in Figure 10, which summarize the results described in section 3.2.2. The criteria of comparing the expert with the algorithm are also highlighted in the description of table 5. Due to the fact that the obtained p-values for the Mann-Whitney U test are very small (close to zero), we decided to leave the p-values in the subfigures (Figures 10), and comparing them to the selected significance level in each individual plot.